# The Effect of Aerobic Training on Healthy Small Airways—A Forced Oscillation Technique Approach to Optimize Long Term Care in COPD

**DOI:** 10.3390/jcm14134755

**Published:** 2025-07-04

**Authors:** Ioan Emanuel Stavarache, Tudor Andrei Cernomaz, Ionela Alina Grosu-Creangă, Antigona Trofor

**Affiliations:** 1Discipline of Pneumology, III-rd Medical Department, Faculty of Medicine, “Grigore T. Popa” University of Medicine and Pharmacy, 700115 Iasi, Romania; stavarash@yahoo.com (I.E.S.); ionela.grossu@yahoo.com (I.A.G.-C.); antigona.trofor@umfiasi.ro (A.T.); 2Department of Preclinical Disciplines, Faculty of Medicine, “Apollonia” University of Iaşi, 700511 Iași, Romania; 3Clinical Hospital of Pulmonary Diseases Iași, 700116 Iasi, Romania

**Keywords:** pulmonary rehabilitation, impulse oscillometry, airway resistance, aerobic training

## Abstract

Limited data exist on the underlying physiological phenomena of aerobic training; the impulse oscillometry method, allowing the assessment of small airways and lung periphery in addition to standard lung function testing, might be a useful addition to rehabilitation programs. **Background/Objectives:** This study aimed to determine the immediate effect of a structured low-intensity aerobic training program on small airway function in healthy volunteers to explore potential implications for long-term COPD care. **Methods:** Thirty-six healthy volunteers were recruited between May 2024 and January 2025; each participant underwent a lung function testing session, followed by low/moderate-intensity aerobic exercise, and, after 15 min, by a second impulse oscillometry assessment. **Results:** There was a statistically significant reduction in airway resistance following the physical exertion for the whole group (mean difference 0.03 kPa/L/s, 95%CI 0–0.6 kPa/L/s); significantly lower values were recorded for the reactance component X5 (0.02 kPa/L/s, 95%CI 0–0.4 kPa/L/s) for the normal weight subgoup (*n* = 24). These results, corroborated with literature data, suggest optimization of the distribution of the airflow and possibly alteration of the elastic properties of the thoracic structures following even low-intensity effort. **Conclusions:** Low-intensity upper body strength and aerobic training seem to have an immediate respiratory beneficial effect on healthy volunteers manifested as a reduction in airway resistance. The underlying mechanism might be related to improved contractility of respiratory muscles, but changes in lung parenchyma elasticity may also be involved, possibly reflecting modifications of ventilation heterogeneity. Impulse oscillometry may be superior to spirometry in monitoring the effects of aerobic training, considering the additional data it provides, and could be used to optimize and personalize rehabilitation protocols.

## 1. Introduction

Chronic obstructive pulmonary disease (COPD) is a leading cause of morbidity and mortality worldwide, imposing a substantial burden on healthcare systems and affecting millions of patients. Current management guidelines emphasize a combination of pharmacotherapy, pulmonary rehabilitation, and lifestyle modifications [1]. While pharmacological interventions remain central to management, non-pharmacological strategies, particularly pulmonary rehabilitation, have gained ground as data emerged showing a positive impact on quality of life, symptom burden, effort capacity, and potentially overall survival [2]. Despite extensive research aiming to optimize pulmonary rehabilitation measures, there is still limited data concerning the objective functional impact, the underlying physiological mechanisms, or the optimal approach to the COPD patient. Furthermore, prescribing and conducting physical exercise for COPD patients may prove difficult as dyspnea, cardiovascular comorbidities, or muscle wasting may prevent high-intensity training; various adverse events, such as variable degrees of desaturation or onset of arrhythmias, may also preclude sustained exercises. Although the effect of aerobic training is generally considered beneficial for either stable disease or following a COPD exacerbation [3], there are pending questions on the optimum delivery or intensity [4]. Available data seem to favor high-intensity regimens, but such approaches may not be acceptable, feasible, or safe for advanced COPD patients [5] with multiple comorbidities; on the other hand, low-intensity exercise still has a positive health impact. Taking this data into account, we aimed to test whether low-intensity aerobic training with resistance training elements has objective lung function effects.

Impulse oscillometry (IOS), a variant of the forced oscillation technique (FOT), bears some advantages over classical lung function testing, placing great emphasis on spirometry. COPD is associated with significant small airway impact—a respiratory component that is not easily characterized in terms of spirometry. Body plethysmography may complement spirometry to determine additional parameters such as airway resistance (Raw) but requires heavier and more expensive equipment and perfect patient collaboration. Some of these disadvantages may be mitigated by using the IOS method, which has two major advantages: easy to perform with high reproducibility and sheds light on various respiratory phenomena missed by classical lung function testing, such as small airway obstruction, compliance or ventilation heterogeneity (a relatively recent concept underlining non-uniform distribution of the air in the lungs typically associated with obstructive respiratory disorders) [6]. Some oscillometry parameters have similar significance to spirometry data, but there is an increasing body of evidence suggesting a better diagnostic sensitivity, at least when COPD underlying physiological changes are concerned [7]; thus, such an approach may provide more comprehensive data compared to standard lung function testing. Impulse oscillometry may complement classic lung function testing as it detects or predicts the existence of an obstructive disorder in patients not able to perform spirometry either due to clinical status [8] or inability to cooperate [9]; furthermore, it offers some insight into small airways physiology and related phenomena [10].

The integration of aerobic training and FOT-based assessments into long-term COPD management strategies may yield several benefits. First, by establishing baseline IOS parameters and tracking changes periodically, clinicians can identify which patients are responders to exercise-based rehabilitation and tailor individualized programs to maximize outcomes. Second, as oscillometry measurements are easy to perform, reproducible, and minimally invasive, they can be conducted at regular intervals, enabling proactive interventions before significant clinical deterioration.

Our study aimed to identify any immediate ventilatory changes following a series of physical exercises classified as low-intensity aerobic training using the impulse oscillometry method. Since this was deemed exploratory, our effort focused on healthy volunteers in order to develop a therapeutic strategy for COPD patients while avoiding unnecessary risks.

## 2. Materials and Methods

### 2.1. Participants

We recruited 36 healthy volunteers between May 2024 and January 2025 from the university population using local posters; there were two inclusion criteria: the absence of any known significant medical condition (respiratory, cardiovascular, or musculoskeletal) and the willingness to participate expressed as signing an informed consent form. There were no exclusion criteria or filters in place with respect to age, gender, or general fitness of the volunteers.

### 2.2. Intervention

All participants underwent lung function testing—both standard spirometry and FOT; then a physical exercise session started under the supervision of a trained physical therapist (details below); the participants were asked to grade the effort intensity on a 1 to 10 visual analog scale; after a 15 min break the FOT was repeated. The exercises were chosen and conducted to count as low intensity (peak heart rate less than 60% of the maximum calculated as 208 − 0.7 × age beats/minute). The visual analog was used to quantify the subjective perception and to confirm that no adverse events were present. Both spirometry and FOT were conducted according to ERS (European Respiratory Society) quality standards regarding technique and calibration using a Vyntus^®^ IOS Vyaire Medical (Mettawa, IL, USA) platform. There was only one case excluded as spirometry showed a moderate obstructive ventilatory deficit—a total of 35 records were analyzed. There was no sample size calculation performed as no previous data on the magnitude of the effect was available to the best of our knowledge.

There were no significant adverse events reported following the physical exercise used as an intervention.

Data collected included—demographics (age, gender), anthropometrics (height, weight), the first second of forced expiration (FEV1) and forced vital capacity (FVC) values, the respiratory resistance (R5Hz), the lung reactance (X5Hz), peripheral airway resistance (D5-20%), the reactance area (AX), the resonant frequency (Fres.) values both pre and after physical exertion and a self-reported score for the intensity of the effort involved.

Statistical analysis was performed using IBM SPSS 26; data was presented as mean ± standard deviation for continuous variables and frequencies for categorical variables. Continuous variables differences were analyzed using either paired or independent samples *t* test (as applicable) using bootstrap resampling. No correction methods were implemented although multiple comparisons were performed, considering the exploratory nature of our study and small sample type II errors were deemed more important and also taking into account the fact that the five variables followed were unrelated.

The set of physical exercises performed consisted of three exercise types repeated 10 times:Standing trunk rotations holding baton with synchronized profound respiratory effortsChest expansion exercises with 0.5 kg dumbbells (lateral lifts)Kettlebell Halo rotation exercise

These exercises were selected aiming to engage the following muscular groups [11]: intercostal, diaphragm, major and minor pectorals, deltoid, oblique abdominal, and erector spinae.

## 3. Results

A total of 35 participants completed both physical exercise and pre- and post-lung function testing; the participant pool composition in terms of age, gender, and body mass index are shown in Table 1—there was a certain gender imbalance and a tendency toward a younger age distribution. The values of R5, X5, D5-20%, AX, and Fres before and after effort were compared using a paired samples *t*-test—Table 2; there were statistically significant lower R5 values after the training series.

All participants rated the effort as low or medium—the distribution of effort intensity self-assessment is shown in Figure 1.

Among the participant pool, we identified two subgroups: normal weight and overweight/obese, using the threshold value of 24 for the body mass index (BMI)—24 participants (68.6%) were deemed to have normal weight. A post hoc analysis showed that X5 was significantly lower following physical exercise only for the normal weight subgroup—Table 3.

## 4. Discussion

### 4.1. Main Findings

Our results suggest that a short series of low-intensity physical aerobic exercises focused on thoracic muscular groups has an immediate effect of lowering the total airway resistance—measured by the impulse oscillometry; there are also some reactance differences—lower X5 values—a difference statistically significant in the normal weight subgroup. These findings challenge traditional assumptions in pulmonary rehabilitation [12]—such as short-term exercise training having little or no effect on ventilatory and transfer parameters. Still, the IOS technique is currently considered to be a robust method focusing on different parameters; furthermore, there is data that suggests a lower variability when compared with standard lung function testing, at least for COPD patients [13]. Seen in this light, we consider the changes we found to reflect a real physiological phenomenon, the magnitude of which may or may not have significant clinical implications.

### 4.2. Ventilatory Parameters

Airway resistance is defined from a mechanistic perspective as the ratio between transpulmonary pressure (calculated as the difference between mouth and alveolar pressures) and the resulting airflow and has a dependence on the radius and length of the airway and the viscosity of the fluid as reflected by the Hagen–Poiseuille equation. Following low-intensity exercise, we cannot infer a significant change in length or diameter of the large airways; the only significant changes might reflect changes in the small airways or thoracic wall elastic properties [14], changes in transpulmonary pressures might also be considered possibly following muscular changes (warm-up) or even more complicated mechanisms such as recruiting and changes in the residual functional capacity.

The R5 component is considered similar to the total airway resistance determined by other measurement methods; the R20 component reflects the large airway component (mainly extra thoracic and central segments); the R5-R20 parameter is considered to reflect the small airway contribution to the total flow resistance. The lack of a significant R5-R20 difference might indicate that the post-effort reduction in total airflow resistance is not explained by a small airway diameter change.

The reactive components dubbed with X reflect the superposition of two-time displaced phenomena provoked by the airflow: an inertial component preceding the airflow and an elastic one following the airflow. When considering the 5 Hz value, the predominant phenomenon is linked to the elasticity of the respiratory system (which explains the negative value) [15].

Our findings of lower airflow resistance following exercise are similar to a murine model study results (Wistar rats with hepatopulmonary syndrome) [16], where moderate aerobic training was associated with a decrease in lung tissue elastance and increased maximal running capacity, a potential role of IL-10 was advanced as a possible explanation. Our data is somewhat similar, but the short time elapsed between the physical effort and the respiratory changes reasonably excludes an inflammatory explanation, the mechanical/muscular mechanism being more probable.

The AX (area of reactance) value and the resonant frequency Fres (corresponding to the point where the reactance spectrum crosses the frequency abscissa) are generally considered to indirectly characterize the lung periphery, considered to be linked to small airways’ permeability and lung compliance. Given their geometrical definitions, AX, Fres, and X5 should be discussed together as they do show a degree of interplay; the post-effort X5 significant decrease with similar AX and Fres values is suggestive of a change in the shape of the negative part (elastance related) of the reactance spectrum. In light of these considerations, the X5 difference might be interpreted as distal airways and lung parenchyma elastance changes; similar results of lower X5 scores were reported as being associated with lung allograft dysfunction reflecting changes in the elasticity of lung parenchyma [17].

### 4.3. Potential Physiological Mechanisms

The exact underlying mechanism of the ventilatory changes we found is probably complex; looking at similar data might provide potential explanations.

For COPD patients, there are published results on thoracoabdominal asynchrony [18] as being associated with respiratory features of end-stage COPD such as dyspnea or lung function parameters—this data might imply that simply correcting posture or training a certain type of breathing might have an objective and measurable respiratory effect.

Similarly, Taichi—a traditional Chinese exercise form bordering martial arts frequently performed as a light daily exercise was reported to have similar results to standard pulmonary rehabilitation after twelve weeks in light of St. Georges Respiratory Questionnaire and effort capacity without a significant spirometry change. This study was started aiming to provide COPD patients with an alternative to classic rehabilitation since it counts as a low to moderate-energy exercise and can be practiced alone in many variants. The underlying mechanism of these improvements was not elucidated [19].

The physical exercises we used in our study are similar in scope to Taichi as they are based on low-intensity stretches mobilizing large groups of thoracic and limb muscles, which may also play an accessory respiratory role; our data may explain such results which include better physical performance without spirometry improvement.

A Chinese randomized controlled study [20] showed lung function improvement—expressed as FEV1/FVC and MVV following the performance of Daoyin exercises in a young population diagnosed with upper crossed syndrome. There are two similarities with our data—the Daoyin techniques are similar in scope and intensity with the exercises we used, and the populations under study are similar; while our study group was considered healthy—anomalies such as upper crossed syndrome were not actively searched and might have been present in our study group given the high prevalence of such conditions amongst young student populations.

A small study that compared 21 controls and 21 intervention participants selected among young sedentary subjects [21] undergoing resistance training consisting of physical exercises similar to the one we used (chest press, dumbbell pullover, flat bench dumbbell fly) repeated 3 times/week for 8 weeks showed improvements in the 6 min walk test score, PEF, FVC, FEV1, MVV, and chest movement ampliation. These results seem to validate our findings, albeit at a different timescale, and also support a musculoskeletal explanation for the multidimensional respiratory improvement.

Similar approaches using variants of resistance training were deemed effective in terms of functional performance with an underlining muscular mechanism for elderly women [22]; the exercises consisted of bench press, deadlift, unilateral rowing, standing calf raise, and lower abdominal exercise.

A small study that included 28 participants reported a significant effect of accessory respiratory muscular groups such as erector spinae, upper trapezius, pectoralis major, sternocleidomastoid, and suboccipitalis on ventilatory parameters in healthy persons maintaining sedentary postures [23] due to working conditions. Similarly, chronic neck pain patients were reported as having lower inspiratory and expiratory pressures, probably reflecting a respiratory muscle weakness mechanism; the authors advance the hypothesis of local and global muscular impairment and also advance the possibility of a psychological component [24]. A more complete image must take into account the results of a small study looking at the relationship between various respiratory parameters and multiple musculoskeletal measurements aimed at assessing the range of motion of the cervical spine and chest mobility; the results showed a relationship between the mobility of the thoracic spine and the chest mobility and the maximal voluntary ventilation without a relationship to the respiratory pressures while suggesting a link between expiratory maximal pressure and the endurance of the neck flexor muscles synergy [25].

### 4.4. Post Hoc Analysis

The reactance changes we found showed up when we performed a post hoc subgroup analysis and seemed connected to the presence of a high BMI value, a parameter that might offer some insight into the underlying physiological mechanism.

There is data pertaining to peripheral lung involvement of obesity in both asthmatic [26] and non-asthmatic overweight patients; weight loss at 12 months after bariatric surgery was associated with modified pulmonary elastance. This elastance change completes the picture of obesity being associated with lower values for frequently used parameters such as FEV1, FVC, PEF, Raw, or R5-R20 [27] in bronchial asthma patients [28]. Along the same line, a study making use of the IOS technique in order to assess the impact of obesity on bronchial asthma advanced a possible association between peripheral airway reactivity and the presence of obesity [29].

A study highlighting differences in ventilatory parameters and changes following physical exertion in normal weight versus overweight versus obese participants found a significant reduction in expiratory time in the obese group [30]. While the underlying mechanism was presumed to be complex and not elucidated by the authors, altered distensibility of the parietal thoracic and abdominal structures was advanced.

The X5 is difficult to interpret alone as increasingly negative values might be seen in both obstructive and restrictive disorders; there are published results suggesting a negative prognostic role for COPD patients [31]. While our data does not allow a definite conclusion, we may postulate a change in ventilatory heterogeneity or a change in the elastic properties of the thoracic structures to explain the difference specifically for normal-weight participants. Considering the limits of a post hoc analysis, we may hypothesize that body weight may play a role in the physiological response to physical effort; therefore, further research to identify maximum impact exercises and to develop protocols to maximize effect should probably take this variable into account.

### 4.5. Extrapolating Results to a COPD-Centered Rehabilitation Program

Exercise training is considered to be central to pulmonary rehabilitation [32], and a wide array of procedures and protocols have been developed. Still, COPD patients may be confronted with various specific difficulties when attempting such exercises—usually stemming from various respiratory and cardiovascular limitations. Muscular atrophy/dysfunction is also prevalent among advanced-stage COPD patients; nutritional and mental disorders may also limit the applicability of exercise training. Taking into account these limitations, developing simple exercises (easy to learn and perform) that do not need dedicated infrastructure and do not place a great burden on the cardiovascular system might be particularly useful for this patient category.

The beneficial role of physical exertion has been proved for COPD patients, at least when larger periods of time are considered; a 6790 participants study reported a protective effect of moderate and intense physical activity on the ventilatory function expressed as FEV1 and FVC after a 10-year follow up period [33]. This effect was evident even for active smokers, and the authors assume an anti-inflammatory role of physical exertion, although the exact mechanism remains unclear.

Work of breath increases with age—even more so in COPD patients—there is data linking this phenomenon not only to increased passive flow resistance but also to an elastic component, at least for the inspiratory phase of ventilation [34]. Furthermore, a study that included 112 stable COPD patients, mainly males, found a negative correlation between the FEV1 value (expressed as a percentage of a predicted value) and various IOS parameters, which included [35].

### 4.6. Limits and Caveats

The present study has some clear limits: the study group was relatively small, gender imbalance was present, a non-standardized set of exercises, and timing of lung function testing. Given the exploratory nature of the study, no sample estimation was conducted prior to enrollment as the magnitude of the effect was unknown; a post hoc analysis shows a power of 50%. Testing only healthy participants somewhat limits the extrapolation of these results to respiratory patients; this is particularly important to consider when COPD patients are involved—as this category might benefit the most from such approaches when considering that COPD is mainly characterized by airflow limitation characterized by increased airflow resistance and ventilatory heterogeneity and hyperinflations frequently following an air trapping mechanism. Some aspects still need to be clarified: the effect duration and the optimum type of exercise (movement, number of repetitions, effort).

## 5. Conclusions

Low-intensity upper body strength and aerobic training seem to have an immediate respiratory beneficial effect on healthy volunteers, manifested as a reduction in airway resistance. The underlying mechanism might be related to improved contractility of respiratory muscles, but changes in lung parenchyma elasticity may also be involved, possibly reflecting modifications in ventilation heterogeneity. Impulse oscillometry may be superior to spirometry in monitoring the effects of aerobic training, considering the additional data it provides, and could be used to optimize and personalize rehabilitation protocols.

## Figures and Tables

**Figure 1 jcm-14-04755-f001:**
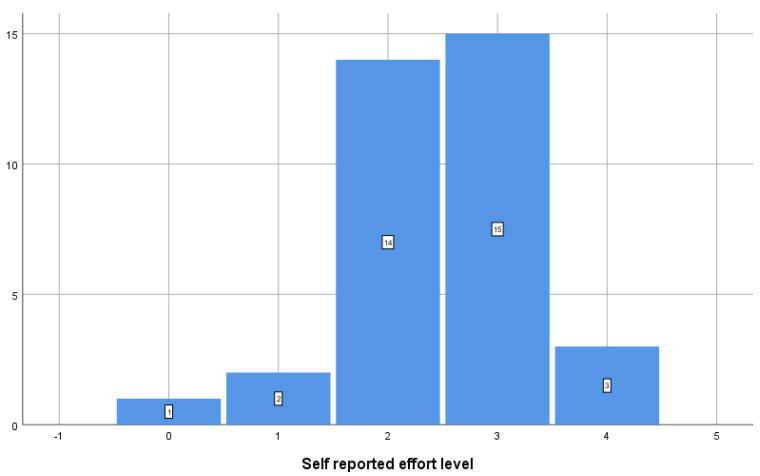
Self-reported effort level using a visual analog questionnaire.

**Table 1 jcm-14-04755-t001:** Participants pool structure—demographics.

	Minimum	Maximum	Mean	Std. Deviation
Age (years)	20	63	32.4	10.5
Body mass index	16	31	24.2	3.3
Gender	Number	Percentage		
Female	9	25.7%		
Male	26	74.2%		

**Table 2 jcm-14-04755-t002:** Main impulse oscillometry parameters at base line and after aerobic training—whole group.

	Mean	Std. Deviation	Mean Difference	95% Confidence Interval of the Difference		*p* Value
R5Hz initial (kPa/L/s)	0.27	0.10	0.03	0.00	0.06	0.05
R5Hz post effort (kPa/L/s)	0.24	0.07				
X5Hz initial (kPa/L/s)	−0.06	0.04	−0.01	−0.05	0.03	0.60
X5Hz post effort (kPa/L/s)	−0.05	0.09				
Fres initial (1/s)	10.51	3.55	0.11	−0.71	0.93	0.79
Fres post effort (1/s)	10.40	3.21				
AX initial (kPa/L)	0.21	0.25	0.02	−0.03	0.08	0.36
AX post effort (kPa/L)	0.19	0.18				
D5-20% initial (%)	7.62	8.59	0.84	−2.04	3.71	0.56
D5-20% post effort (%)	6.78	7.41				

**Table 3 jcm-14-04755-t003:** Main impulse oscillometry parameters at baseline and after aerobic training—normal weight subgroup (*n* = 24).

	Mean	Std. Deviation	Mean Difference	95% Confidence Interval of the Difference		*p* Value
R5Hz initial (kPa/L/s)	0.25	0.10	0.02	−0.02	0.06	0.26
R5Hz post effort (kPa/L/s)	0.23	0.07				
X5Hz initial (kPa/L/s)	−0.04	0.04	0.02	0.00	0.04	0.04
X5Hz post effort (kPa/L/s)	−0.06	0.03				
Fres initial (1/s)	9.44	3.05	0.00	−0.57	0.58	0.99
Fres post effort (1/s)	9.43	2.64				
AX initial (kPa/L)	0.17	0.23	0.01	−0.05	0.06	0.84
AX post effort (kPa/L)	0.16	0.15				
D5-20% initial (%)	4.85	7.08	−0.12	−2.67	2.43	0.92
D5-20% post effort (%)	4.96	6.83				

## Data Availability

The original contributions presented in this study are included in the article. Further inquiries can be directed to the corresponding author(s).

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
