# Peer review of "The Effect of Aerobic Training on Healthy Small Airways—A Forced Oscillation Technique Approach to Optimize Long Term Care in COPD"

_jcm, 2025, doi:10.3390/jcm14134755_

Round 1
Reviewer 1 Report
Comments and Suggestions for Authors
Dear Author,
Thank you for your well-structured and comprehensive study, "The Effect of Aerobic Training on Healthy Small Airways – A Forced Oscillation Technique Approach to Optimize Long-Term Care in COPD." Your manuscript is clinically relevant and provides valuable insights into the potential role of aerobic training in small airway function and COPD management.
Please find detailed comments attached in the review report. To further enhance the quality of your manuscript, I strongly recommend professional English proofreading to eliminate grammatical errors and spelling mistakes, ensuring improved readability, clarity, and academic precision.
Additionally, I suggest structuring the Discussion section into subsections with clear subheadings to improve logical flow and reader engagement. This will help effectively present key findings, physiological interpretations, comparisons with existing literature, and clinical implications.
I appreciate your effort in refining the manuscript and look forward to your revised submission.
Best regards,

To further enhance the quality of your manuscript, I strongly recommend professional English proofreading to eliminate grammatical errors and spelling mistakes, ensuring improved readability, clarity, and academic precision.
Reviewer 2 Report
Comments and Suggestions for Authors
Thank you for the opportunity to review this paper. The authors assess the effect of single low/moderate intensity aerobic exercise on IOS parameters. The topic is interesting and using IOS for evaluation adds to the existing literature on its place for clinical practice.
Still some major issues exist. I think the main one is the fact that the authors assess IOS after only a single session, and even then, without any assessment of longer-term effect. In addition, I cannot understand how can they extrapolate their results to COPD patients. This might not have the same results in patients with COPD. Why not including patients with COPD to understand its effect on COPD? The fact that no additional assessments were done to evaluate the effect of such intervention and that the change in IOS parameters is clinically relevant. Finally, a significant limitation is the lack of general novelty of this trial. Other studies have used IOS pre and post exercise to evaluate small airways and the literature on the effect of exercise on COPD (including of pulmonary rehabilitation) is very wide, and this study does not really add a lot to it.
Other major limitations are the small sample size and limited effect size, with only R5 being slightly different. The fact that small airways parameters (AX and R5-20) are similar from pre to post training also weakens any real conclusion from this trial.
Comments on the Quality of English LanguageSee above
Author Response
Thank you very much for the time and effort invested into reviewing our manuscript. Please find below point to point responses to the issues raised in the comments.
Thank you for the opportunity to review this paper. The authors assess the effect of single low/moderate intensity aerobic exercise on IOS parameters. The topic is interesting and using IOS for evaluation adds to the existing literature on its place for clinical practice.
Still some major issues exist. I think the main one is the fact that the authors assess IOS after only a single session, and even then, without any assessment of longer-term effect.
The aim was to assess if a low intensity training session has any quantifiable ventilatory effects - at least detectable by impulse oscillometry.
In addition, I cannot understand how can they extrapolate their results to COPD patients. This might not have the same results in patients with COPD. Why not including patients with COPD to understand its effect on COPD?
This is a proof of concept of a larger program – it felt safer and simpler to test a concept on healthy volunteers than to include COPD patients which may also bring variability in terms of phenotypes and disease stages.
The fact that no additional assessments were done to evaluate the effect of such intervention and that the change in IOS parameters is clinically relevant. Finally, a significant limitation is the lack of general novelty of this trial. Other studies have used IOS pre and post exercise to evaluate small airways and the literature on the effect of exercise on COPD (including of pulmonary rehabilitation) is very wide, and this study does not really add a lot to it.
There are few studies looking at the underlying mechanisms of rapid ventilatory improvement following low intensity training. The majority of published data is based on spirometry or CPET and the results are often inconclusive in terms of strict ventilatory improvements.
Other major limitations are the small sample size and limited effect size, with only R5 being slightly different. The fact that small airways parameters (AX and R5-20) are similar from pre to post training also weakens any real conclusion from this trial.
Being an exploratory study with no previous data to allow sample size calculation the risk of low power was considered and discussed.
The text was modified to better reflect the limitations of the study.
Round 2
Reviewer 1 Report
Comments and Suggestions for Authors
Improved the manuscript
Author Response
N/A
Reviewer 2 Report
Comments and Suggestions for Authors
Thank you for the opportunity to review this paper once again. The authors have addressed each of my comments and I thank them for that.
I understand the study aim and appreciate the use of IOS. Still, there is NO connection between what was done and COPD. How does your work is a proof of concept? Change in IOS following exercise/rehab was performed by multiple studies before (PMID: 35570301, PMID: 35806942, PMID: 36363507, PMID: 38306350). In addition, there is no issue of safety of such an intervention in COPD patients. Many studies have shown that exercise training in COPD is not only safe, by effective and patients could perform such training at home without supervision. This is a heavily investigated area of research and you cannot say "proof of concept" just to justify your sample size and extrapolation to COPD.
Therefore, I cannot see how you can write COPD in the title or have any conclusion on COPD using your paper. The focus on COPD in the abstract, intro and discussion is also not appropriate.
I would suggest a justification for your sample size – i.e. a statistical one.
In addition, I would add in the intro (instead of the data on COPD) the different uses of IOS and prior evidence on its role in assessing airways following acute events or exercise. For example, a prospective study assessed the use of IOS in patients with dyspnea and found it to have a good utility to identify obstructive disease, therefore a good tool to use for patients unable to perform spirometry (DOI: 10.4187/respcare.10963). This and other examples should be given.
Comparisons should not be made by t-tests as I assume the measurements of IOS parameters were not normally distributed considering the small sample size. In addition, all comparisons between pre and post should be of paired samples.
Table 1 should be revised and the title of each column should be changes as the number +percentages of males/females is not a minimum/maximum. This is really a small amount of data (only age, sex and BMI) which could be presented in the text and not in a table.
Figure 1 can also be revised to be more attractive.
Finally, your conclusions do not match the results. Only R5 was different from pre to post in your cohort. This also has only a limited significance and could not be counted as a "beneficial effect". By dividing your cohort, you further minimize your sample size, significantly limiting your results. Other conclusions that are not supported by anything in your results – "Impulse os-cillometry may be superior to spirometry in monitoring", "changes in lung parenchyma elasticity may also be involved possibly reflecting modifications of ventilation heterogeneity", and many more.
I would shorten the discussion, addressing only issues directly assessed by this study.
Another main issue that is not mentioned is the MCID for IOS parameters. This was evaluated by different studies and should be compared to the study results.
Author Response
Thank you for the time and effort put into reviewing the manuscript – please find enclosed the answers to the issues raised.
Thank you for the opportunity to review this paper once again. The authors have addressed each of my comments and I thank them for that.
I understand the study aim and appreciate the use of IOS. Still, there is NO connection between what was done and COPD. How does your work is a proof of concept? Change in IOS following exercise/rehab was performed by multiple studies before (PMID: 35570301, PMID: 35806942, PMID: 36363507, PMID: 38306350)
Response:
PMID: 35570301 12 scleroderma patients – after a 12 week program there is FOT improvement
PMID: 35806942, 48 patients with IIP subjected to a 3-week inpatient PR – no FOT improvement
PMID: 36363507, ~150 patients, three week pulmonary rehabilitation– no FOT improvement
PMID: 38306350) strength and inspiratory muscle training over a period of 8 weeks, 2–3 times a week for 3–4 hours each – IOS found superior to spirometry
The use of impulse oscillometry is not new – however the afore mentioned articles refer to weeks long rehabilitation programs with mixed results. The article aims to characterize immediate physiological changes – and to the best of our knowledge there is not much data published on this topic.
In addition, there is no issue of safety of such an intervention in COPD patients. Many studies have shown that exercise training in COPD is not only safe, by effective and patients could perform such training at home without supervision. This is a heavily investigated area of research and you cannot say "proof of concept" just to justify your sample size and extrapolation to COPD.
Therefore, I cannot see how you can write COPD in the title or have any conclusion on COPD using your paper. The focus on COPD in the abstract, intro and discussion is also not appropriate.
Response
Exercise training is generally deemed safe for COPD patients – still the British Thoracic Society Clinical Statement on pulmonary rehabilitation doi:10.1136/thorax-2023-220439 states:
‘Most validation studies have taken place in clinical settings where the tests were directly
supervised and therefore the safety and validity of remotely supervised functional tests in patients with chronic respiratory disease have not been established.’ – which is within the scope of the article.
Furthermore the same statement mentions among current research gaps:
‘Studies to assess the safety and validity of remotely supervised exercise and functional outcomes through videoconferencing or mobile applications.
Alternative strategies to prescribe exercise and deliver effective PR in the absence of a directly supervised validated exercise test.’
Along the same line both this statement and other guidelines raise concerns about comorbidities (mainly cardiovascular) and recommend personalized programs.
I would suggest a justification for your sample size – i.e. a statistical one.
Response
There is no possible prior calculation for the sample size unless the effect is known and the magnitude may be inferred; both were deemed unknown. The final group size was determined by the number of volunteers who agreed to participate.
In addition, I would add in the intro (instead of the data on COPD) the different uses of IOS and prior evidence on its role in assessing airways following acute events or exercise. For example, a prospective study assessed the use of IOS in patients with dyspnea and found it to have a good utility to identify obstructive disease, therefore a good tool to use for patients unable to perform spirometry (DOI: 10.4187/respcare.10963). This and other examples should be given.
Response
Text was modified
Comparisons should not be made by t-tests as I assume the measurements of IOS parameters were not normally distributed considering the small sample size. In addition, all comparisons between pre and post should be of paired samples.
Response
All tests were paired samples as it was stated in the material and method section; normal distribution assumption was checked using Shapiro Wilk and bootstrapping was used to circumvent normality issues
Table 1 should be revised and the title of each column should be changes as the number +percentages of males/females is not a minimum/maximum. This is really a small amount of data (only age, sex and BMI) which could be presented in the text and not in a table.
Response
Table was modified
Figure 1 can also be revised to be more attractive.
Finally, your conclusions do not match the results. Only R5 was different from pre to post in your cohort. This also has only a limited significance and could not be counted as a "beneficial effect".
Response
There is an improvement of roughly 10% for R5 which is close to RAW – while this is not necessarily spectacular it is not negligible - for example when compared to standard pharmacological therapy for asthma or COPD patients. Jalusic-Gluncic T. What happens with airway resistance (RAW) in asthma and COPD exacerbation. Med Arh. 2011;65(5):270-3. PMID: 22073849.
By dividing your cohort, you further minimize your sample size, significantly limiting your results.
Response
The subgroups were created as part of a post hoc analysis – given the fact that we were dealing with normal subjects. Obesity/overweight status may associate ventilatory defects therefore such a stratification was reasonable.
Other conclusions that are not supported by anything in your results – "Impulse os-cillometry may be superior to spirometry in monitoring", "changes in lung parenchyma elasticity may also be involved possibly reflecting modifications of ventilation heterogeneity", and many more.
I would shorten the discussion, addressing only issues directly assessed by this study.
Response
There is no clear cut published data underlining the utility of spirometry for monitoring rapid lung function changes following pulmonary rehabilitation at least as classic parameters are concerned. Considering this, IOS might be superior as it seems to detect some differences concerning parameters irrelevant to spirometry.
Changes in reactance associated to resistance improvements while R20-R5 is not significantly altered might highlight a ventilation heterogeneity mechanism rather than a small airways one as it is presented in the discussion section.
There are no many more conclusions.
Another main issue that is not mentioned is the MCID for IOS parameters. This was evaluated by different studies and should be compared to the study results.
Response
Available MCIDs for IOS parameters are disease and sometimes tool specific – there is no consensus for healthy subjects.